# How repeated exposure to informal science education affects content knowledge of and perspectives on science among incarcerated adults

Joshua J. Horns *, Nalini Nadkarni, Allison Anholt

School of Biological Sciences, University of Utah, Salt Lake City, Utah, United States of America

* joshua.horns@utah.edu

## Abstract

Over two million men, women, and youth are incarcerated in the United States. This large and ethnically diverse population has, in general, more limited exposure to education, particularly in scientific fields, than the general public. Formal educational programs for the incarcerated can be expensive and logistically difficult to initiate and maintain, but informal science education (ISE) approaches have the potential to significantly improve inmates' view of science and of themselves as science learners. However, "dosage effects"—how repeated exposure to educational experiences may affect listeners–are poorly documented. In this study, we evaluated the longitudinal effects of an ISE program in Utah, which provided a monthly lecture series delivered by academic scientists on a range of science topics. Science content knowledge, self-perception as a science learner, interest in science, and a desire to seek out more scientific information all significantly improved for inmates attending lectures. We also found that seeing a greater number of lectures is positively associated with a desire to seek out additional information. We documented an inverse relationship between education background and the increase in a desire to learn more, suggesting that those with more limited exposure to science manifest the greatest increase in seeking out more information. These results suggest that ISE for the incarcerated significantly improves their knowledge of, and relationship with, science; that some of these effects carry over across months or years; and that ISE programs can have the largest impact by focusing on those with more limited prior exposure to science.

## Introduction

Public communication and public engagement are increasingly becoming priorities for individual scientists and scientific agencies, and the scientific enterprise [1–4]. The highest priority lies in programs that engage populations that have been traditionally underserved by science. One of the largest and most underserved populations in the Unites States are the 2.1 million incarcerated adults and 50,000 incarcerated youth in prisons, jails, and juvenile detention

**Data Availability Statement:** All relevant data are within the paper and its Supporting Information files.

**Funding:** Funding for this project was generously funded to NN by an anonymous donor (and thus we are unable to supply a grant number, name, or URL). The funders had no role in study design, data collection and analysis, decision to publish, or preparation of the manuscript.

**Competing interests:** The authors have declared that no competing interests exist.

centers [5–7]. There are multiple compelling reasons why public engagement programs should focus on the incarcerated. First, although educational backgrounds vary substantially, on average, people incarcerated in the criminal justice system have more limited academic background and quality of education than the general population [8]. Second, exposure to education while incarcerated reduces recidivism, increases the probability of post-release employment, translates into reductions in crime, taxpayer savings, and provides greater safety in communities to which formerly incarcerated people return [9, 10].

The number of formal educational programs in prisons in growing [11, 12]. However formal programs require significant time and financial resources to establish and maintain. The majority of such programs focus their limited resources on basic and secondary education, life skills, vocational training aimed at post-release employment, and functional social behavior [8, 13, 14]. Most formerly incarcerated students in higher education gravitate toward political science, economics, and other disciplines in the social sciences [15]. An alternative approach is the implementation of informal science education (ISE) programs in correctional facilities. These programs offer lectures or workshops to help inmates gain inspiration and knowledge that may promote a successful return to their communities [16, 17] but generally offer no academic credit or certificate. ISE has many benefits in a correctional settings, including low overhead, less required structure, and less reliance on inmate continuity (many of whom have short sentences or are unpredictably transferred among different cellblocks or correctional institutions) attending a concurrent series of events.

Since 2014, the Initiative to bring Science Programs to the Incarcerated (INSPIRE) has been delivering monthly ISE lectures at various correctional institutions near Salt Lake City, UT. INSPIRE brings science and nature to the incarcerated and build connections between science, scientists, inmates, and corrections staff through lectures, workshops, and conservation projects. Over 100 scientists from various academic institutions and agencies in Utah have participated in INSPIRE. Each scientist delivered a 45-minute lecture on his/her area of research. Topics did not adhere to any formal curriculum, but rather reflected the research areas and disciplinary fields of the participating scientists (e.g., water quality, human genetics, astrobiology, complete set of lecture topics in S1 Table). Lectures took place at the Draper State Prison and Salt Lake County Jail and have reached over 4,468 inmates. All inmates were given the opportunity to fill out surveys before and after each lecture. Questions were designed to gauge their science-content knowledge, how they view themselves as science-learners, interest in science, and their desire to seek out additional scientific information [18].

A previous study conducted by INSPIRE suggested that in the first year of the program's existence, inmates that attended lectures had significant improvement in their science content knowledge and in how they view science [18]. However, because the program was newly established at the time, researchers were unable to test the longitudinal effects of ISE and see whether these effects persist over a number of years and how inmates' perspectives shift after being exposed to multiple lectures.

This effect of dosage (how repeated exposure to educational experiences may shift lecture attendees' content knowledge of and perspectives on science) has been largely overlooked by correctional educational studies. Dosage effects have been highlighted as an important area of future research in many studies of correctional education [9, 19, 20, 21]. Pedagogical studies of formal education in students ranging from 5th grade through college have documented that repeated exposure to a topic increases students' recall of fundamental principles [22, 23] and reduces the impacts of periods of standard academic decline [24]. The few studies that have investigated coarse-scale differences in education dosage amongst the incarcerated (i.e., high school diploma vs. vocational training; working toward a GED vs. earning a GED) have found that more exposure leads to lower changes of recidivism and better post-release employment

[19, 21, 25]. However, still unknown is how repeated exposure to informal science education while incarcerated may impact how inmates gain science content knowledge, view science, and see themselves as science-learners. This information is important in prioritizing the allocations of limited correctional education resources.

In this study, we examined the effects of ISE at one correctional institution over a five-year period. We used mixed-effects ordinal regression to investigate how science content knowledge, self-perception as science-learners, interest in science, and a desire to seek out additional information change before and after a lecture, after seeing multiple lectures, and by demographics. Based on earlier research [18], we hypothesized that responses would be significantly more positive in post-lecture surveys compared to pre-lecture surveys and that this effect would be consistent across years. We further hypothesized that responses would improve with repeated exposure to science lectures. Lastly, we predicted that inmates with more limited prior exposure to education will exhibit the greatest improvement in science content knowledge and attitudes regarding science. Inmates with poorer educational backgrounds might manifest the greatest improvement in science knowledge content because those with more previous exposure to scientific concepts would more likely correctly answer knowledge questions before seeing a lecture and therefore have less potential for improvement, whereas those that have had more limited exposure to science would be more likely to answer incorrectly. Inmates with less educational background might likewise see the largest improvement in attitudes towards science since unfamiliarity with a topic tends to be associated with less interest [26, 27].

## Methods

This research was conducted in accordance with all relevant human-subjects guidelines and approved by the University of Utah's Internal Review Board (IRB_00061095). Consent was obtained from each participant through a written agreement distributed before the start of each lecture detailing the information to be collected and the purpose. Inmates who did not give consent were still welcome to attend lectures. Although INSPIRE operates at multiple institutions, we focused our analyses on inmates at Draper State Prison, where inmates have the longest sentences, the least amount of turnover and therefore the opportunity to see a greater number of lectures over more years. The prison is managed by the Utah Department of Corrections and has a capacity of 4,300 inmates from minimum to maximum security. Lectures were given to men in medium-security participating in substance-abuse or sex-offender treatment ("Promontory" cellblock). Attendance at lectures was voluntary. Approximately one lecture each month was given at the prison beginning in January 2014. The program is on-going, and as of August 2019 had presented 54 lectures at Draper State Prison. All lectures were delivered by volunteer scientists and topics varied from month-to-month depending on the lecturer's area of research (S1 Table). Lectures were typically 45 minutes with a 15-minute question-and-answer period immediately after. A set of custom readings on the topic of the lecture (10 pages maximum) was compiled for each lecture, and made available at the end of each lecture to supplement the content of the lecture.

Attendees were given pre- and post-lecture surveys to complete with 18 (pre) to 31 (post) questions. Questions were primarily five-point Likert scale with some short answers. All Likert-scale questions fell into one of four main categories: (1) Science content knowledge, three true/false questions specific to the topic of the day's lecture (SC); (2) Self-perception as a science learner (SP); (3) Interest in science (IS), and; (4) Behavioral intention, i.e., how likely the respondent was to seek out or discuss science information on their own time (BI). Survey questions were not entirely consistent across years, however. Therefore, we focused our

analyses on lectures from June 2016 onward since which time questions remained unchanged excluding three lectures which reverted to an alternate set of questions (S1 Table). This resulted in data from 34 lectures. A complete list of questions included in analyses are in S2 Table. Responses were on a 1 (strongly disagree/very unlikely) to 5 (strongly agree/very likely) scale. This pattern was inverted for two questions with reversed polarity ("Only highly trained scientists can understand science" and "Scientific work would be too hard for me"). Responses were likewise inverted for content questions in which the statement was false (e.g., "Fruit flies are not useful for genetic research").

To analyze how responses compared before and after lectures, after seeing additional numbers of lectures, and by demographics, we ran cumulative link mixed models for each of the four categories of questions. Dependent variables were the 1–5 ordinal responses from the surveys. Fixed effects included number of lectures seen, whether the answers were pre- or post-lecture, and highest level of education. Participant ID was included as a random effect. We included interaction terms between all three fixed effects based on our hypothesis that the differences between pre- and post-lecture answers would change based on the educational background of the inmate and the number of previous lectures they had attended. Although we only included responses from the lectures with consistent questions (beginning in June 2016), we calculated the number of lectures an inmate had attended across all years. Education was ranked based on traditional pattern of education progression (i.e., some college > high school diploma > some high school). Schooling that did not fall into traditional progression (e.g., trade school) was considered intermediate between levels of traditional education (thus a response of "some high school with trade school" was ranked above "some high school" alone but below "high school diploma"). All analyses were performed in R [28], cumulative link mixed models were conducted using package 'ordinal' [29].

## Results

Across all 34 lectures delivered since June 2016, attendees completed and returned 862 pre-surveys and 719 post-surveys spread across 475 unique offenders. 662 surveys had matching pre and post surveys completed by the same individual during a single lecture. The majority of respondents (n = 300, 63%) attended only one lecture and the most lectures attended (by a single inmate) was 18 (Fig 1A). The most common highest-level of education was a high school diploma (n = 157, 33%) but educational backgrounds ranged from junior high or less (n = 7, 1%) to graduate school (n = 13, 3%) (Fig 1B).

Pre-lecture scores for science content knowledge were roughly evenly split, with 53% of inmates reporting the correct answer. However, pre-lecture scores for self-perception, interest in science, and behavioral intention were overwhelmingly positive with 81%, 81%, and 75% answering positively respectively, suggesting an audience eager for more STEM education.

In all four categories of question, responses were always significantly higher after the lecture than before ($z_{sc}$ = 6.14 ($p < 0.001$), $z_{sp}$ = 2.57 ($p = 0.010$), $z_{is}$ = 2.41 ($p = 0.016$), $z_{bi}$ = 138.92 ($p < 0.001$), Fig 2). Likewise, inmates with greater educational backgrounds had significantly more positive answers for all question categories ($z_{sc}$ = 3.12 ($p = 0.0018$), $z_{sp}$ = 4.481 ($p < 0.001$), $z_{is}$ = 2.61 ($p = 0.009$), $z_{bi}$ = 35.23 ($p < 0.001$)).

We found no significant interactions between any of our cofactors for interest in science and self-perception categories. However, for behavioral intention, we found strong inverse interactions between the differences in pre- vs. post-lecture score and both the number of lectures attended ($z = -16.317$ ($p < 0.001$), Fig 3) and educational background ($z = -6.716$ ($p < 0.001$), Fig 4) suggesting that a person's desire to seek out additional information after a

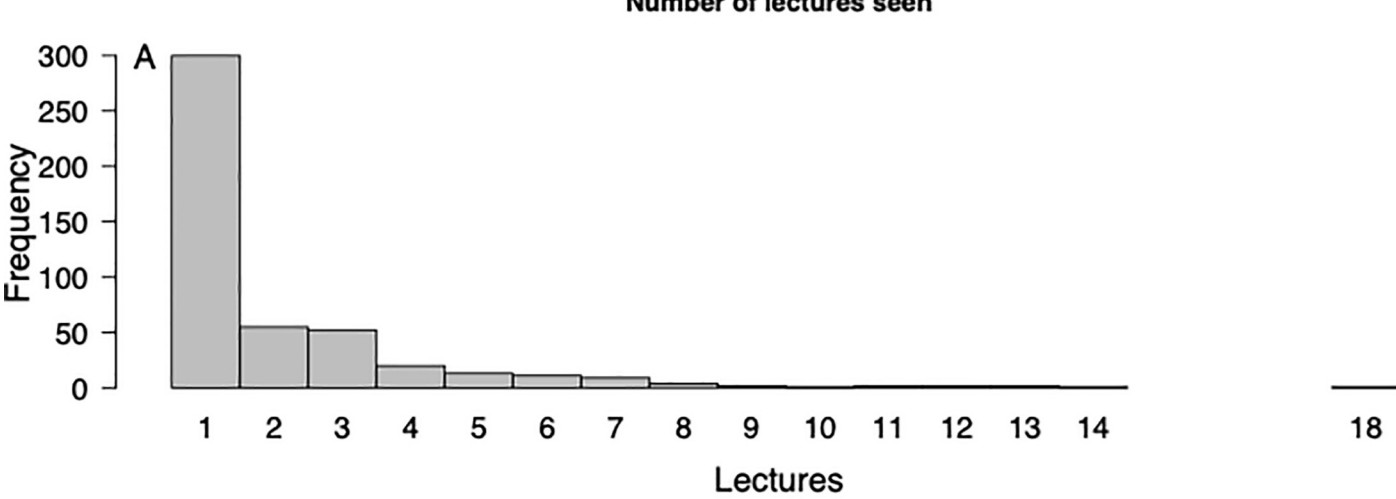

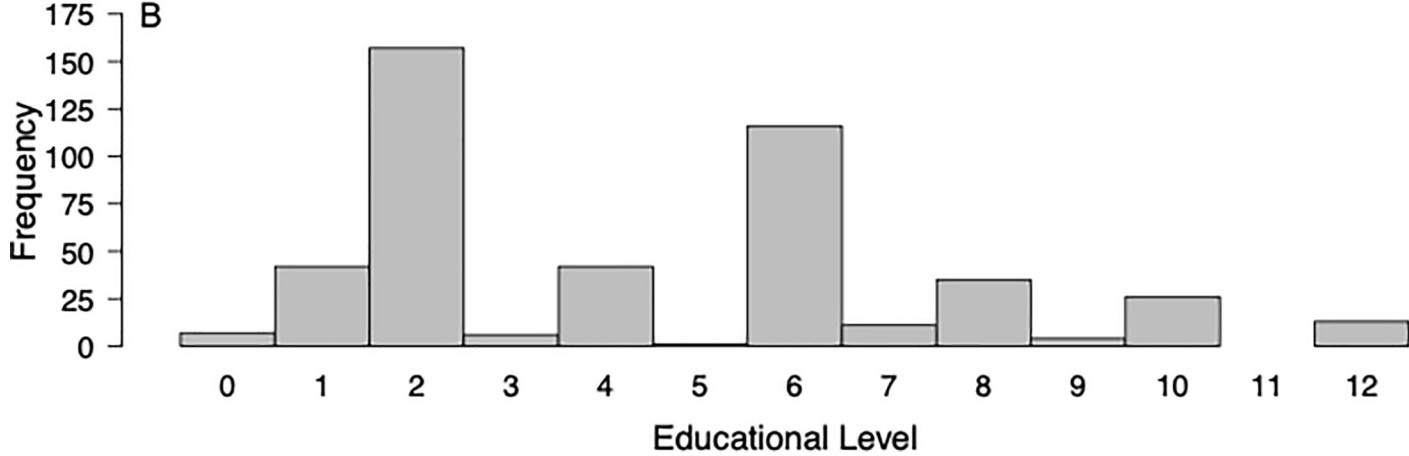

**Fig 1.** Histograms of the number of lecture attendants broken down by number of lectures seen (A) and highest level of educational attainment (B).

lecture improves the greatest in those newer to the program and those with a more limited history of exposure to education.

Number of lectures seen played a significant role for behavioral intention where inmates that had seen more lectures were more likely to want to seek out additional scientific information ($z = 27.49$ ($p < 0.001$), Fig 3). Combined with the inverse interaction between number of lectures attended and difference in pre- vs. post-lecture responses, this suggests that repeated exposure significantly increases attendees' desire in finding additional information and that this effect carries over between lectures with inmates attending subsequent lectures with a higher *a priori* interest (Fig 3).

Contrary to our expectations, we found a positive (rather than negative) interaction between the difference in pre- vs. post-lecture score and educational background for science content knowledge suggesting that inmates with greater educational exposure retain more content ($z = 2.33$ ($p = 0.020$), Fig 4).

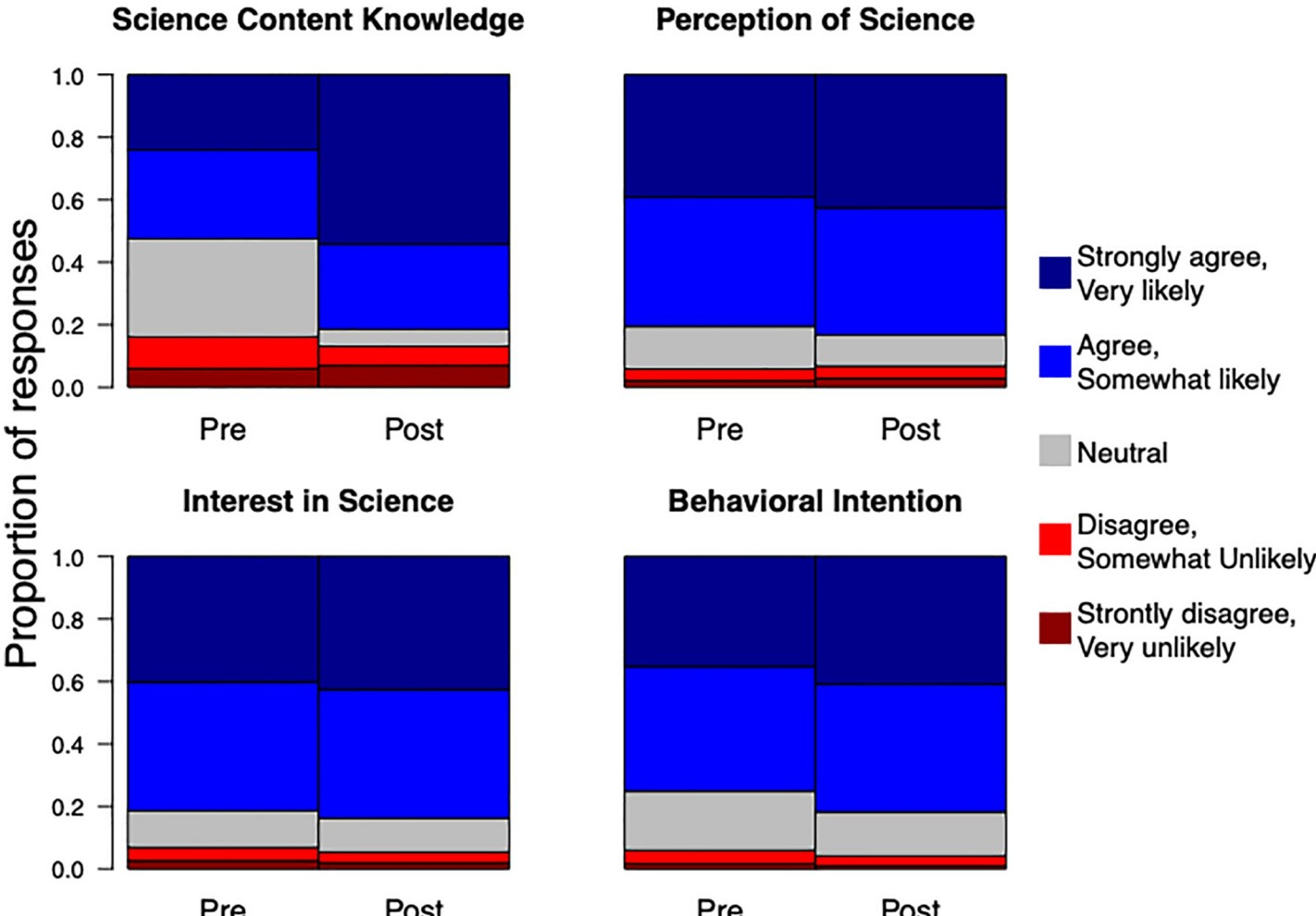

**Fig 2. Proportion of inmates responding from 1 (strongly disagree, very unlikely) to 5 (strongly agree, very likely) in four categories of matching pre/post survey questions.** Responses were inverted for questions with reversed polarity. Mixed-effect ordinal regression identified responses as significantly more positive in all categories. Science-content knowledge questions, three true/false questions specific to each lecture, were standardized in analyses so that "strongly agree" was always the correct response.

## Discussion

Results from inmates surveyed immediately before and after attending STEM lectures over a five-year period suggested that science content knowledge, self-perception as a science-learner, interest in science, and desire to learn more all significantly increased after seeing a lecture across the course of our study. One question category, interest in seeking out additional information, was consistent with our prediction in that scores continually improved with repeated exposure to STEM lectures. Also as predicted, the greatest improvement in scores was among inmates with more limited educational backgrounds and those that were newer to the program. Contrary to our expectations, inmates with more prior exposure to education saw the most improvement in science content scores possibly due to more experience taking in and retaining information.

Based on a previous study that surveyed incarcerated populations before and after attending STEM lectures [18], we anticipated positive improvement in pre- vs. post-lecture scores. However, the fact that this pattern was largely consistent regardless of the number of lectures an

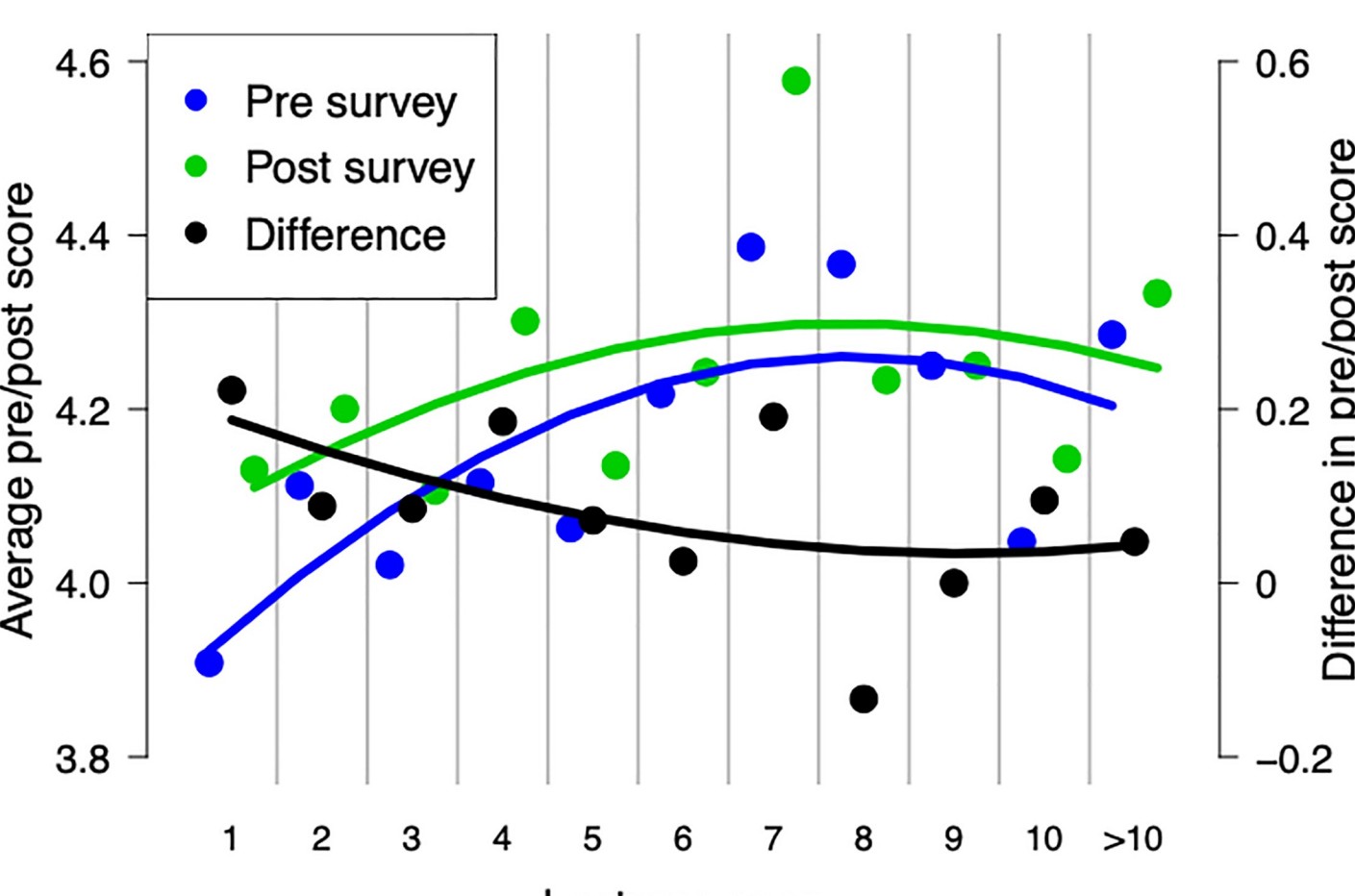

**Fig 3. Average responses to behavior intention questions before (blue) and after (green) attending different numbers of lectures.** Black points represent the difference between post- and pre-lecture responses. Lines are smoothed regressions. Number of lectures significantly improved behavioral intention scores in mixed-effect ordinal regression. The difference in pre/post lecture scores was significantly greater in inmates who had attended fewer numbers of lectures. Responses from inmates that had attended over 10 lectures were lumped for visualization but left unaltered in models.

inmate had previously attended suggests that there are compelling reasons to maintain ISE programs over long periods of time. For some of our questions, which showed no significant association with dosage (e.g., self-perception as a science-learner and interest in science,) it may be that such perceptions are difficult to permanently alter using ISE and that participants require regular infusion of STEM education to revitalize their interest and inspire confidence. Perceptions of science as "too difficult" have been suggested as the main reason why students opt not to pursue STEM careers [27, 30]. Thus, a program that can break down these preconceptions for inmates may help improve both attitudes towards science as well as post-release employment prospects.

We found that behavioral intention (BI) scores, (i.e., the likeliness that an inmate will seek out additional information or discuss science with others), continually improved with attendance at additional lectures regardless of whether the responses were pre- or post-lecture. This may be due to topics more familiar to a respondent often coming across as more interesting [31]. If repeated exposure to STEM topics increases inmate familiarity with science in general,

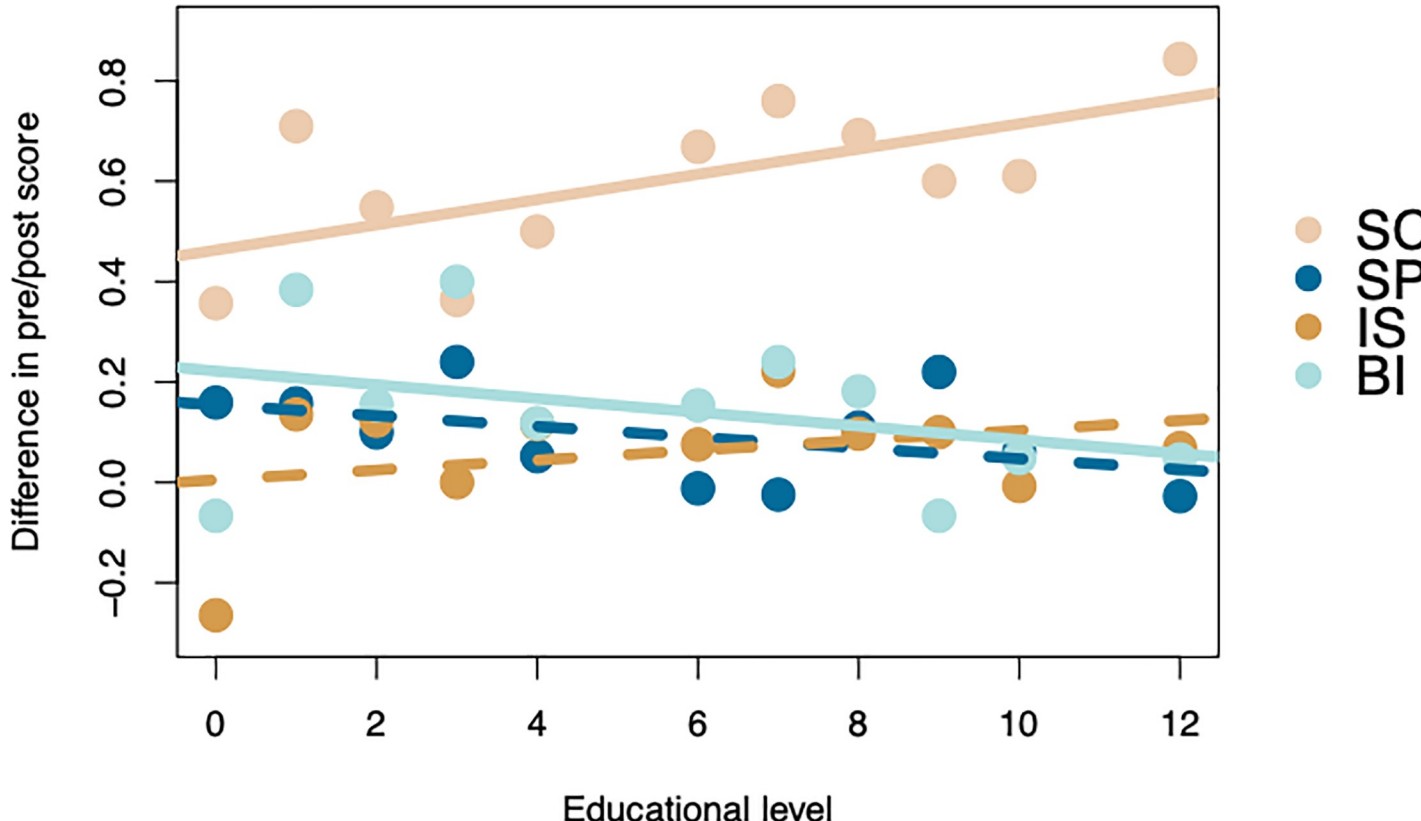

**Fig 4. The difference in post- vs. pre-lecture score in all four categories of survey question: Science content knowledge (SC), self-perception as a science learner (SP), interest in science (IS), and behavioral intention (BI).** Solid lines represent relationships found to be significant in mixed-effect ordinal regression, dashed lines are insignificant patterns. The significant positive interaction between education and pre/post score for science content suggest that those with higher educational backgrounds retain more information about the lecture. The significant negative interaction between education and pre/post score for behavioral intention suggests that inmates new to the program see the largest increase in their desire to seek out additional information.

it may lead to a greater degree of interest. This result is particularly interesting when set alongside the inverse relationship between pre/post BI scores and number of lectures attended. This suggests that, although post-lecture BI scores are consistently higher than pre-lecture scores, the difference becomes smaller for inmates who have attended a greater number of lectures (Fig 3 (or 4)). Combined with the positive effect that lecture number had on BI scores, our findings suggest that both pre- and post-lecture scores continually increase with more lectures attended but that the rate of this increase is greatest in pre-lecture scores. This means that inmates may approach subsequent lectures already more interested to seek out and discuss information related to science.

As may have been expected, inmates with greater educational backgrounds scored consistently higher on science content knowledge (SC) questions potentially because they had prior exposure to the topics being discussed. However, contrary to our expectations, inmates with greater educational backgrounds also showed the greatest improvement in pre- vs. post-lecture SC scores. We had hypothesized that those with less previous exposure to science education may be most unfamiliar with science-specific questions before a lecture and therefore have the greatest potential for improvement. The fact that we saw the opposite pattern may be due to inmates with more exposure to science, and education in general, possessing greater background knowledge and working memory within which new information is processed and stored [32, 33].

This result contrasts with pre/post BI scores which improved the greatest in inmates with less previous education. Previous exposure to science has been identified as a strong positive influence in students' attitudes towards science [26, 27]. Thus, the larger improvement in BI scores for inmates with more limited prior education may be caused by their exposure to science through INSPIRE lectures. Importantly for this type of program, research suggests that positive shifts in attitudes towards science are possible through extra-curricular ISE in addition to more traditional STEM education [26].

The inmates' presence at INSPIRE lectures was voluntary and therefore inmates that chose to attend may be those with a favorable predisposition towards science and education, potentially biasing results. However, attending inmates represented a diverse range of prior educational experience (40% of respondents had either failed to graduate, or had no education beyond, high school) and pre-survey attitude responses of first-time attendees, though positively skewed, still contained over 1,500 responses (nearly ¼ of the total) that were negative or neutral.

## Conclusions

Surveys of inmates before and after attending STEM lectures indicated that science content knowledge, self-perception as science-capable, interest in science, and curiosity in learning more were all significantly improved by participating in lectures. Moreover, we found that an interest to learn more continually increased the more lectures an inmate attended and that those with less formal education saw the greatest improvement. These results provide strong support for the need to establish and maintain long-term ISE programs in correctional institutions. ISE is a cost-effective means of "de-mystifying" science for the incarcerated, particularly for those with less formal education, improving attitudes towards science, scientists, and their own perception as being science-capable.

## Supporting information

**S1 Table. A list of lecture topics given at Draper State Prison.** Lectures in bold were those with consistent pre/post questions included in analyses.
(DOCX)

**S2 Table. Likert-scale questions used in mixed-effects modelling.** All questions had five possible responses from strongly agree to strongly disagree (A/D) or very likely to very unlikely (L/U). Science content questions were specific to the topic of each lecture.
(DOCX)

**S1 Data.**
(CSV)

## Acknowledgments

We acknowledge the collaboration of the Utah Department of Corrections and the University of Utah College of Science. We thank Jeremy Morris, Matthew Whittaker, Stacy Eddings, Program Director Victor Kersey, and Megan Young for their help. The University of Utah's Institutional Review Board (IRB_00061095) provided oversight and Human Subjects Review for activities.

## Author Contributions

**Conceptualization:** Joshua J. Horns, Nalini Nadkarni.

**Data curation:** Joshua J. Horns.

**Formal analysis:** Joshua J. Horns.

**Funding acquisition:** Nalini Nadkarni.

**Investigation:** Nalini Nadkarni.

**Methodology:** Joshua J. Horns.

**Project administration:** Nalini Nadkarni, Allison Anholt.

**Resources:** Nalini Nadkarni, Allison Anholt.

**Supervision:** Nalini Nadkarni.

**Visualization:** Joshua J. Horns.

**Writing – original draft:** Joshua J. Horns, Nalini Nadkarni.

**Writing – review & editing:** Joshua J. Horns, Nalini Nadkarni, Allison Anholt.

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
