## [Decision Letter · Decision Letter 0]

29 Apr 2020

How repeated exposure to informal science education affects content knowledge of and perspectives on science among incarcerated adults

PONE-D-19-29261

Dear Dr. Horns,

We are pleased to inform you that your manuscript has been judged scientifically suitable for publication and will be formally accepted for publication once it complies with all outstanding technical requirements.

With kind regards,

Slavko Rogan

Academic Editor

PLOS ONE

Journal Requirements:

2. Please provide additional information regarding the consent considerations made for the prisoners included in this study. In the methods section please provide more detail on how consent was obtained and include a copy of the consent agreement or verbal script used as an "Other" file. Additonally, please clarify whether individuals who did not participate received the same treatment offered to participants (could they attend lectures without opting into the study).

Reviewers' comments:

Reviewer's Responses to Questions

**Comments to the Author**

1. Is the manuscript technically sound, and do the data support the conclusions?

Reviewer #1: Yes

Reviewer #2: Yes

2. Has the statistical analysis been performed appropriately and rigorously? 

Reviewer #1: Yes

Reviewer #2: I Don't Know

3. Have the authors made all data underlying the findings in their manuscript fully available?

Reviewer #1: Yes

Reviewer #2: Yes

4. Is the manuscript presented in an intelligible fashion and written in standard English?

Reviewer #1: Yes

Reviewer #2: Yes

5. Review Comments to the Author

Reviewer #1: These findings contribute to our understanding of dosage for education programs offered to incarcerated individuals. The methods and analytic approach were sound. Expose to STEM courses offers an avenue of exposure I hadn't considered before. it would be interesting in the future if possible to do randomization.

Reviewer #2: Overall this paper is very well written and addresses a topic that is often overlooked. I am very pleased that the authors have conducted this research and made these data available to the public. The only aspects that I believe could have been mentioned would be the possible implications and applications of this study, i.e. have there been any studies on the effects of science learning and identity as a science learner on a prisoners' rehabilitation and re-entrance into society?

6. PLOS authors have the option to publish the peer review history of their article (what does this mean?). If published, this will include your full peer review and any attached files.

Reviewer #1: Yes: Lois M. Davis, Ph.D.

Reviewer #2: Yes: Elizabeth Marie Watts

---

## [Editor Report · Acceptance letter]

13 May 2020

PONE-D-19-29261 

How repeated exposure to informal science education affects content knowledge of and perspectives on science among incarcerated adults 

Dear Dr. Horns:

I am pleased to inform you that your manuscript has been deemed suitable for publication in PLOS ONE. Congratulations! Your manuscript is now with our production department. 

With kind regards,

on behalf of

Dr. Slavko Rogan 

Academic Editor

PLOS ONE